# Differential temporal dynamics during visual imagery and perception

**Nadine Dijkstra\*, Pim Mostert, Floris P de Lange, Sander Bosch, Marcel AJ van Gerven**

Donders Institute for Brain, Cognition and Behaviour, Radboud University, Nijmegen, Netherlands

**Abstract** Visual perception and imagery rely on similar representations in the visual cortex. During perception, visual activity is characterized by distinct processing stages, but the temporal dynamics underlying imagery remain unclear. Here, we investigated the dynamics of visual imagery in human participants using magnetoencephalography. Firstly, we show that, compared to perception, imagery decoding becomes significant later and representations at the start of imagery already overlap with later time points. This suggests that during imagery, the entire visual representation is activated at once or that there are large differences in the timing of imagery between trials. Secondly, we found consistent overlap between imagery and perceptual processing around 160 ms and from 300 ms after stimulus onset. This indicates that the N170 gets reactivated during imagery and that imagery does not rely on early perceptual representations. Together, these results provide important insights for our understanding of the neural mechanisms of visual imagery.

DOI: https://doi.org/10.7554/eLife.33904.001

\*For correspondence:
n.dijkstra@donders.ru.nl

Visual imagery is the ability to generate visual experience in the absence of the associated sensory input. This ability plays an important role in various cognitive processes such as (working) memory, spatial navigation, mental rotation, and reasoning about future events (*Kosslyn et al., 2001*). When we engage in visual imagery, a large network covering parietal, frontal and occipital areas becomes active (*Ganis et al., 2004*; *Ishai et al., 2000*). Multivariate fMRI studies have shown that imagery activates similar distributed representations in the visual cortex as perception for the same content (*Reddy et al., 2010*; *Lee et al., 2012*; *Albers et al., 2013*). There is a gradient in this representational overlap, in which higher, anterior visual areas show more overlap between imagery and perception than lower, posterior visual areas (*Lee et al., 2012*; *Horikawa and Kamitani, 2017*). The overlap in low-level visual areas furthermore depends on the amount of visual detail required by the task (*Kosslyn and Thompson, 2003*; *Bergmann et al., 2016*) and the experienced imagery vividness (*Albers et al., 2013*; *Dijkstra et al., 2017a*). Thus, much is known about the spatial features of neural representations underlying imagery. However, the temporal features of these representations remain unclear.

In contrast, the temporal properties of perceptual processing are well studied. Perception is a highly dynamic process during which representations change rapidly over time before arriving at a stable percept. After signals from the retina reach the cortex, activation progresses up the visual hierarchy starting at primary, posterior visual areas and then spreading towards secondary, more anterior visual areas over time (*Serre et al., 2007*; *Lerner et al., 2001*; *Van Essen et al., 1992*). First, simple features such as orientation and spatial frequency are processed in posterior visual areas (*Hubel and Wiesel, 1968*) after which more complex features such as shape and eventually semantic category are processed more anteriorly (*Maunsell and Newsome, 1987*; *Vogels and Orban, 1996*; *Seeliger et al., 2017*). After this initial feedforward sweep, feedback from anterior to

**eLife digest** If someone stops you in the street to ask for directions, you might find yourself picturing a particular crossing in your mind's eye as you explain the route. This ability to mentally visualize things that we cannot see is known as visual imagery. Neuroscientists have shown that imagining an object activates some of the same brain regions as looking at that object. But do these regions also become active in the same order when we imagine rather than perceive?

Our ability to see the world around us depends on light bouncing off objects and entering the eye, which converts it into electrical signals. These signals travel to an area at the back of the brain that processes basic visual features, such as lines and angles. The electrical activity then spreads forward through the brain toward other visual areas, which perform more complex processing. Within a few hundred milliseconds of light entering the eye, the brain generates a percept of the object in front of us.

So, does the brain perform these same steps when we mentally visualize an object? Dijkstra et al. measured brain activity in healthy volunteers while they either imagined faces and houses, or looked at pictures of them. Electrical activity spread from visual areas at the back of the brain to visual areas nearer the front as the volunteers looked at the pictures. But this did not happen when the volunteers imagined the faces and houses. Contrary to perception, the different brain areas did not seem to become activated in any apparent order. Instead, the brain areas active during imagining were those that only became active during perception after 130 milliseconds. This is the time at which brain areas responsible for complex visual processing become active when we look at objects.

These findings shed new light on how we see with our mind's eye. They suggest that when we imagine an object, the brain activates the entire representation of that object at once rather than building it up in steps. Understanding how the brain forms a mental image in real time could help us develop new technologies, such as brain-computer interfaces. These devices aim to interpret patterns of brain activity and display the output on a computer. Such equipment could help people with paralysis to communicate.

DOI: https://doi.org/10.7554/eLife.33904.002

posterior areas is believed to further sharpen the visual representation over time until a stable percept is achieved (*Cauchoix et al., 2014*; *Bastos et al., 2015*; *Bastos et al., 2012*).

However, the temporal dynamics of visual imagery remain unclear. During imagery, there is no bottom-up sensory input. Instead, visual areas are assumed to be activated by top-down connections from fronto-parietal areas (*Mechelli et al., 2004*; *Dijkstra et al., 2017b*). Given the absence of bottom-up input, it might be that during imagery, high-level representations in anterior visual areas are activated first, after which activity is propagated down to more low-level areas to fill in the visual details. This would be in line with the reverse hierarchy theory (*Ahissar and Hochstein, 2004*; *Hochstein and Ahissar, 2002*). Alternatively, there may be no ordering such that during imagery, perceptual representations at different levels of the hierarchy are reinstated simultaneously. Furthermore, the temporal profile of the overlap between imagery and perception remains unclear. The finding that imagery relies on similar representations in low-level visual cortex as perception (*Albers et al., 2013*) suggests that imagery might already show overlap with the earliest stages of perceptual processing. On the other hand, it is also possible that more processing is necessary before the perceptual representation has a format that can be accessed by imagery, which would be reflected by overlap arising later in time.

In the current study, we investigated these questions by tracking the neural representations of imagined and perceived stimuli over time. We combined magnetoencephalography (MEG) with multivariate decoding. First, we investigated the temporal dynamics within perception and imagery by exploring the stability and recurrence of activation patterns over time (*King and Dehaene, 2014*; *Astrand et al., 2015*). Second, we investigated the temporal overlap by exploring which time points during perception generalized to imagery.

# Results

## Behavioral results

Twenty-five participants executed a retro-cue task in which they perceived and imagined faces and houses and rated their experienced imagery vividness on each trial using a sliding bar (see *Figure 1*). Prior to scanning, participants filled in the Vividness of Visual Imagery Questionnaire, which is a measure of people's imagery ability (*Marks, 1973*). There was a significant correlation between VVIQ and averaged vividness ratings ($r = -0.45$, p=0.02), which indicates that people with a higher imagery vividness as measured by the VVIQ also rated their imagery as more vivid on average during the experiment. Participants reported relatively high vividness on average (49.6 ± 26.6 on a scale between −150 and +150, where 0 is the starting point of the slider and positive numbers indicate high vividness and negative numbers indicate low vividness). There was no significant difference in vividness ratings between faces (54.0 ± 29.7) and houses (48.7 ± 26.7; $t(24)$ = 1.46, p=0.16). To ensure that participants were imagining the correct images, on 7% of the trials participants had to indicate which of four exemplars they imagined. The imagined exemplar was correctly identified in 89.8% (±5.4%) of the catch trials, indicating that participants performed the task correctly. There was also no significant difference between the two stimulus categories in the percentage of correct catch trials (faces: 90.9 ± 6.6, houses: 88.8 ± 7.1; $t(24)$ = −1.25, p=0.22). Furthermore, correctly identified catch trials were experienced as more vivid (52.06 ± 33.1) than incorrectly identified catch trials

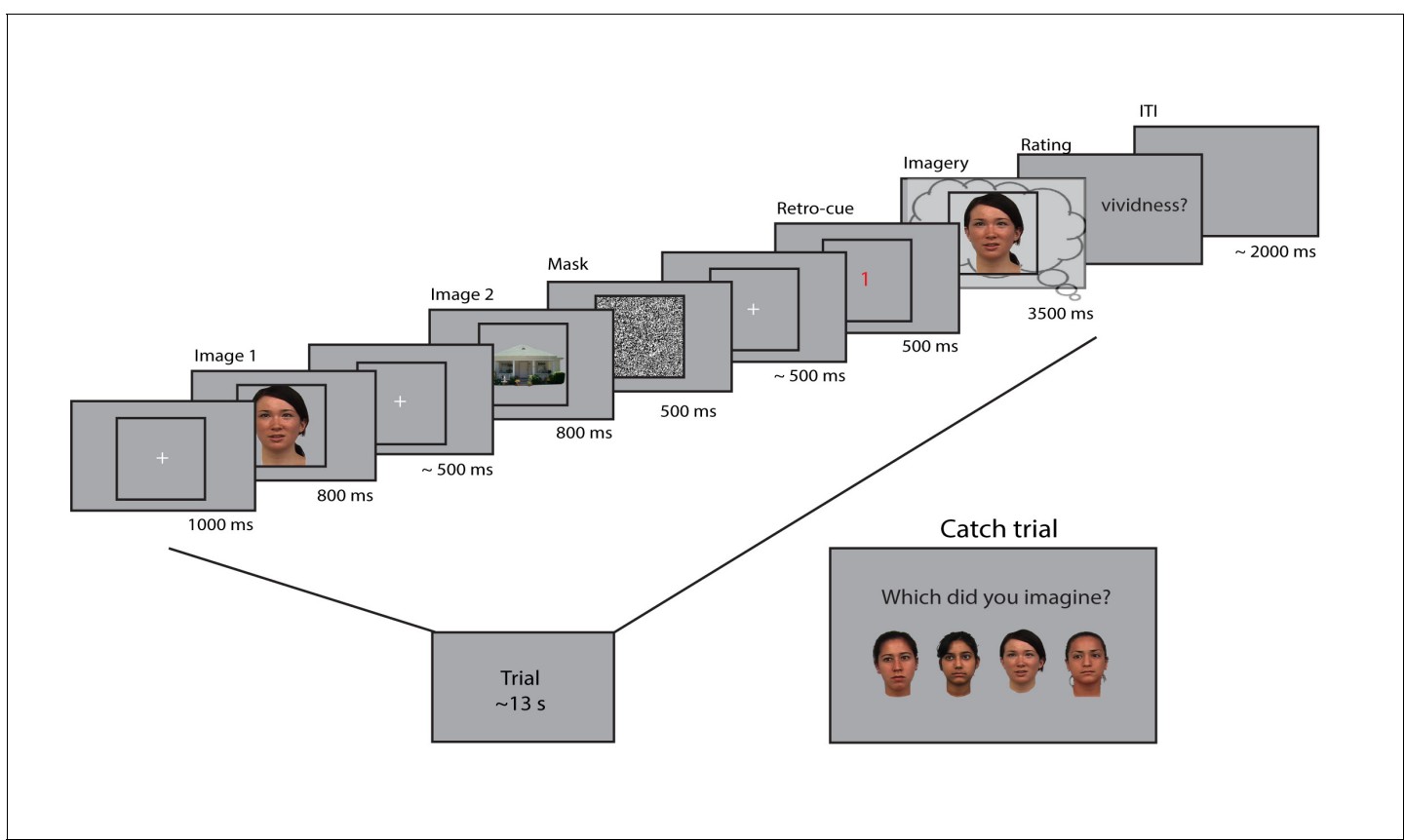

**Figure 1.** Experimental design. Two images were presented for 0.8 seconds each, with a random inter-stimulus interval (ISI) between 400 and 600 ms. After the second image, a mask with random noise was on screen for 500 ms. The retro-cue indicating which of the two images the participants had to imagine was shown for 500 ms. Subsequently, a frame was presented for 3.5 s within which the participants imagined the cued stimulus. After this, they rated their experienced vividness on a continuous scale. On a random subset (7%) of trials, the participants indicated which of four exemplars they imagined that trial. The face stimuli were adapted from the multiracial face database (courtesy of Michael J Tarr, Center for the Neural Basis of Cognition and Department of Psychology, Carnegie Mellon University (http://www.tarrlab.org) and available at http://wiki.cnbc.cmu.edu/Face_Place under a Creative Commons Attribution-NonCommercial-ShareAlike 3.0 Unported License (https://creativecommons.org/licenses/by-nc-sa/3.0/)).
DOI: https://doi.org/10.7554/eLife.33904.003

(24.31 ± 33.84; *t*(23) = 3.11, p=0.0049), giving support to the criterion validity of the identification task. We had to drop one subject for this comparison because this person did not have any incorrect trials.

## Representational dynamics during perception and imagery

To uncover the temporal dynamics of category representations during perception and imagery, we decoded the category from the MEG signal over time. The results are shown in *Figure 2*. Testing and training on the same time points revealed that during perception, significantly different patterns of activity for faces and houses were present from 73 ms after stimulus onset with the peak accuracy at 153 ms (*Figure 2A*, left). During imagery, category information could be decoded significantly from 540 ms after retro-cue onset, with the peak at 1073 ms (*Figure 2A*, right, *Figure 2—figure supplement 1B*). The generation of a visual representation from a cue thus seems to take longer than the activation via bottom-up sensory input. Furthermore, imagery decoding resulted in a much lower accuracy than perception decoding (peak perception is at ~90%, peak imagery at ~60%). This is also observed in fMRI decoding studies (*Reddy et al., 2010*; *Lee et al., 2012*) and is probably due to the fact that imagery is a less automatic process than perception, which leads to higher between trial variation in neural activations. Note that, to allow better comparison between perception and imagery, we only showed the first 1000 ms after cue onset during imagery (see *Figure 2—figure supplement 1* for the results throughout the entire imagery period).

To reveal the generalization of representations over time, classifiers were trained on one time point and tested on all other time points (*King and Dehaene, 2014*) (*Figure 2B*). Furthermore, to investigate the temporal specificity of the representations at each time point, we calculated the proportion of off-diagonal classifiers that had a significantly lower accuracy than the diagonal classifier of that time point (*Astrand et al., 2015*) (*Figure 2C*; see Materials and methods).

During perception, distinct processing stages can be distinguished (*Figure 2B–C*, left). During the first stage, between 70 and 120 ms, diagonal decoding was significant and there was very high temporal specificity. This indicates sequential processing with rapidly changing representations (*King and Dehaene, 2014*). During this time period, the classifier mostly relied on activity in posterior visual areas (*Figure 2D*, left). Therefore, these results are consistent with initial feedforward stimulus processing. In the second stage, around 160 ms, the classifier generalized to neighboring points as well as testing points after 250 ms. The associated sources are spread out over the ventral visual stream (*Figure 2D*, left), indicating that high-level representations are activated at this time. In the third stage, around 210 ms, we again observed high temporal specificity (*Figure 2C*, left) and a gap in generalization to 160 ms (*Figure 2B*, left). This pattern could reflect feedback to low-level visual areas. Finally, from 300 ms onwards there is a broad off-diagonal generalization pattern that also generalizes to time points around 160 ms and an associated drop in temporal specificity (*Figure 2B–C*, left). This broad off-diagonal pattern likely reflects stabilization of the visual representation.

In contrast, during imagery, we did not observe any clear distinct processing stages. Instead, there was a broad off-diagonal generalization throughout the entire imagery period (*Figure 2B*, right; *Figure 2—figure supplement 1A*). Already at the onset of imagery decoding, there was high generalization and low specificity (*Figure 2B–C*, right). This indicates that the neural representation during imagery remains highly stable (*King and Dehaene, 2014*). The only change seems to be in decoding strength, which first increases and then decreases over time (Fig. S1B), indicating that either representations at those times are weaker or that they are more variable over trials. The sources that contributed to classification were mostly located in the ventral visual stream and there was also some evidence for frontal and parietal contributions (*Figure 2D*, right).

Even though we attempted to remove eye movements from our data as well as possible (see Materials and methods), it is theoretically possible that eye movements which systematically differed between the conditions caused part of the neural signal that was picked up by the decoding analyses (*Mostert et al., 2017*). In order to investigate this possibility, we tried to decode the stimulus category from the X and Y position of the eyes as measured with an eye tracker. The results for this analysis are shown in *Figure 2—figure supplement 2*. During imagery, eye tracker decoding was at chance level for all time points, indicating that there were no condition-specific eye movements during imagery (*Figure 2—figure supplement 2B*). However, during perception, eye tracker decoding was significant from 316 ms onwards (*Figure 2—figure supplement 2A*), indicating that differences in eye movements between the conditions may have driven (part of) the brain decoding. If this were

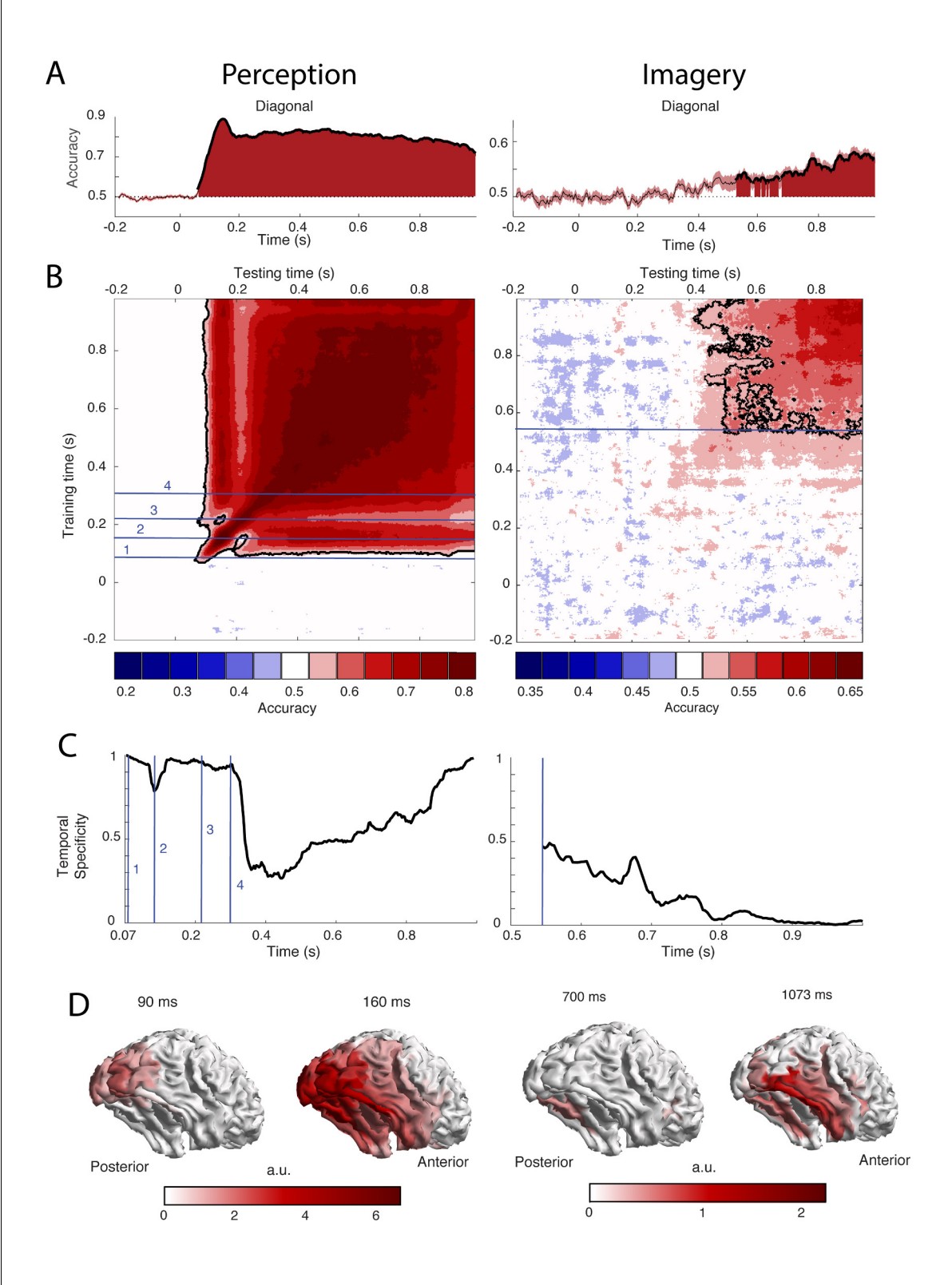

**Figure 2.** Decoding performance of perception and imagery over time. (A) Decoding accuracy from a classifier that was trained and tested on the same time points. Filled areas and thick lines indicate significant above chance decoding (cluster corrected, p<0.05). The shaded area represents the standard error of the mean. The dotted line indicates chance level. For perception, zero signifies the onset of the stimulus, for imagery, zero signifies the onset of the retro-cue. (B) Temporal generalization matrix with discretized accuracy. Training time is shown on the vertical axis and testing time on

*Figure 2 continued on next page*

*Figure 2 continued*

the horizontal axis. Significant clusters are indicated by black contours. Numbers indicate time points of interest that are discussed in the text. (C) Proportion of time points of the significant time window that had significantly lower accuracy than the diagonal, that is specificity of the neural representation at each time point during above chance diagonal decoding (D) Source level contribution to the classifiers at selected training times. Source data for the analyses reported here are available in *Figure 2—source data 1*.

DOI: https://doi.org/10.7554/eLife.33904.004

The following source data and figure supplements are available for figure 2:

**Source data 1.** Temporal dynamics within perception and imagery.

DOI: https://doi.org/10.7554/eLife.33904.008

**Source data 2.** Decoding from eye tracker data.

DOI: https://doi.org/10.7554/eLife.33904.009

**Source data 3.** Vividness median split.

DOI: https://doi.org/10.7554/eLife.33904.010

**Figure supplement 1.** Decoding results throughout the entire imagery period.

DOI: https://doi.org/10.7554/eLife.33904.005

**Figure supplement 2.** Decoding on eye tracker data.

DOI: https://doi.org/10.7554/eLife.33904.006

**Figure supplement 3.** Differences in decoding accuracy between high and low vivid participants during perception and imagery.

DOI: https://doi.org/10.7554/eLife.33904.007

the case, there would be a high, positive correlation between eye tracker decoding and brain decoding. *Figure 2—figure supplement 2C* however shows that there was no such correlation, suggesting that our perception decoding results for that time window were not driven by eye movements.

## Temporal overlap between perception and imagery

To investigate when perceptual processing generalizes to imagery, we trained a classifier on one data segment and tested it on the other segment. We first trained a classifier during perception and then used this classifier to decode the neural signal during imagery (*Figure 3A–B*). Already around 350 ms after imagery cue onset, classifiers trained on perception data from 160, 700 and 960 ms after stimulus onset could significantly decode the imagined stimulus category (*Figure 3A*). This is earlier than classification within imagery, which started at 540 ms after cue onset (*Figure 2A*, right). Considering the increased decoding accuracy during perception compared to imagery (*Figure 2A*, left versus right), this difference might reflect an increase in signal-to-noise ratio (SNR) by training on perception compared to imagery.

Furthermore, the distinct processing stages found during perception (*Figure 2B*, left) were also reflected in the generalization to imagery (*Figure 3A–B*). Perceptual processes around 160 ms and after 300 ms significantly overlapped with imagery (*Figure 3B*, right plots). In contrast, processing at 90 ms did not generalize to any time point during imagery (*Figure 3B*, top left). Perceptual processing at 210 ms showed intermittent generalization to imagery, with generalization at some time points and no generalization at other times (*Figure 3B*, bottom left). Significant generalization at this time could also reflect the effects of smoothing over neighboring time points which are significant (see Materials and methods). This would mean that there is no real overlap at 210 ms but that this overlap is caused by overlap from earlier or later time points.

To further pinpoint when perception started to overlap with imagery, we performed an additional analysis in which we reversed the generalization: we trained classifiers on different time points during imagery and used these to classify perception data. This analysis revealed a similar pattern of high overlap with perception around 160 and after 300 ms and low overlap before 100 ms and around 210 ms (*Figure 3C–D*). Note that this profile is stable throughout imagery and is already present at the start of imagery, albeit with lower accuracies (*Figure 3*-D, bottom panel). Furthermore, the onset of perceptual overlap is highly consistent over the course of imagery: overlap starts around 130 ms, with the first peak at approximately 160 ms (*Figure 3C*). In general, cross-classification accuracy was higher when training on imagery than when training on perception (*Figure 3C* vs. *Figure 3A*). This is surprising, because training on high SNR data (in our case, perception) is reported to lead to higher classification accuracy than training on low SNR data (*King and Dehaene, 2014*) (imagery). This difference may reflect the fact that the perceptual representation contained more unique features than

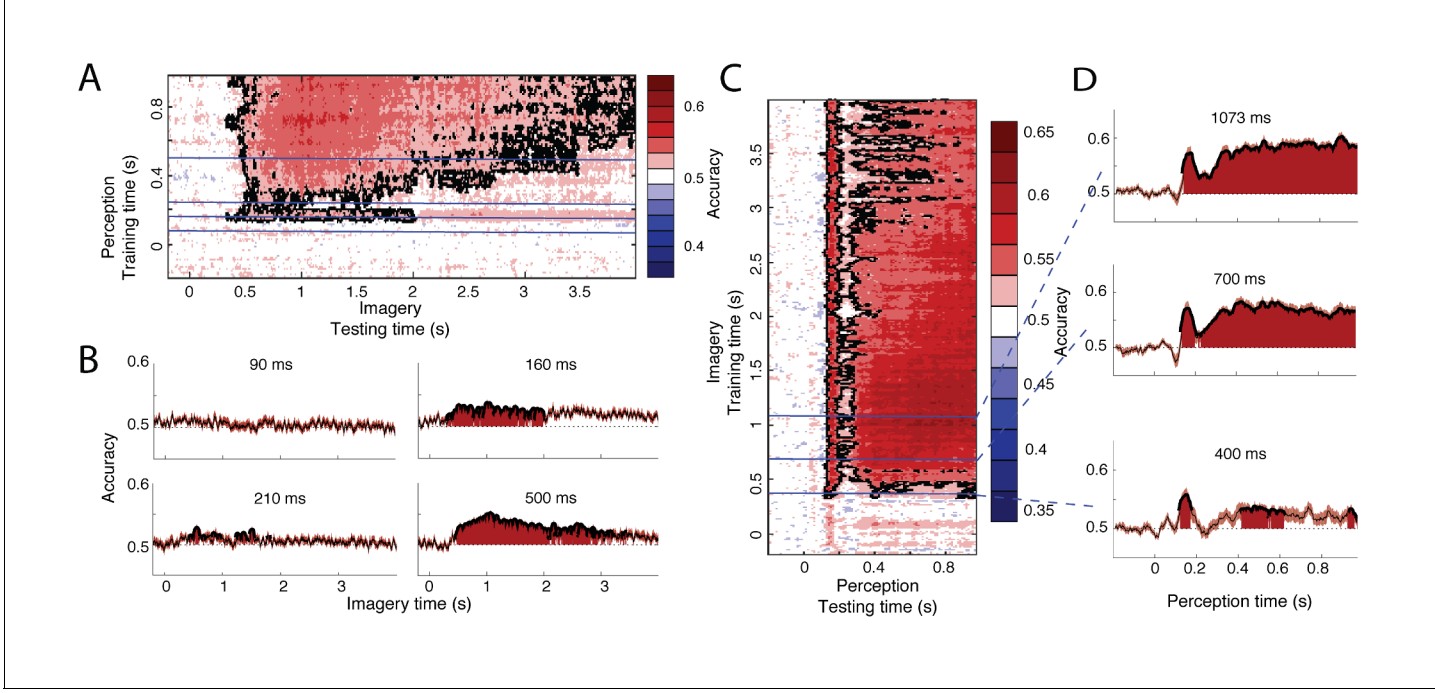

**Figure 3.** Generalization between perception and imagery over time. (**A**) Decoding accuracy from classifiers trained on perception and tested during imagery. The training time during perception is shown on the vertical axis and the testing time during imagery is shown on the horizontal axis. (**B**) Decoding accuracies for classifiers trained on the four stages during perception. (**C**) Decoding accuracy from classifiers trained on imagery and tested during perception. The training time during imagery is shown on the vertical axis and the testing time during perception is shown on the horizontal axis. (**D**) Decoding accuracies for different training times during imagery. Source data for the analyses reported here are available in *Figure 3—source data 1*.

DOI: https://doi.org/10.7554/eLife.33904.011

The following source data and figure supplement are available for figure 3:

**Source data 1.**

DOI: https://doi.org/10.7554/eLife.33904.013

**Source data 2.**

DOI: https://doi.org/10.7554/eLife.33904.014

**Figure supplement 1.** Differences in cross-decoding accuracy between high and low vivid participants.

DOI: https://doi.org/10.7554/eLife.33904.012

the imagery representation, leading to a lower generalization performance when training on perception.

We also investigated whether the temporal dynamics were influenced by imagery vividness by investigating whether the results of previous analyses were different for participants with high or low vividness (see *Figure 2—figure supplement 3* for within perception and imagery decoding and *Figure 3—figure supplement 1* for cross-decoding). Decoding accuracy seemed to be higher in the high vividness group, however, none of the differences were significant after correction for multiple comparisons.

## Discussion

We investigated the temporal dynamics of category representations during perception and imagery, as well as the overlap between the two. We first showed that, compared to perception, imagery decoding became significant later, indicating that it takes longer to generate a visual representation based on purely top-down processes. Furthermore, whereas perception was characterized by high temporal specificity and distinct processing stages, imagery showed wide generalization and low temporal specificity from the onset. Finally, cross-decoding between perception and imagery

revealed a very clear temporal overlap profile which was consistent throughout the imagery period. We observed overlap between imagery and perceptual processing starting around 130 ms, decreasing around 210 ms and increasing again from 300 ms after stimulus onset. This pattern was already present at the onset of imagery.

These findings cannot be explained by a clear cascading of activity up or down the visual hierarchy during imagery. If there was a clear order in activation of different areas, we would not have observed such wide temporal generalization at the start of imagery but instead a more diagonal pattern, as during the start of perception (*King and Dehaene, 2014*). Furthermore, we found that the complete overlap with perception was already present at the onset of imagery.

One interpretation of our results is that during imagery the complete stimulus representation, including different levels of the hierarchy, is activated simultaneously. However, there was no overlap between imagery and perceptual processing until 130 ms after stimulus onset, when the feedforward sweep is presumably completed and high-level categorical information is activated for the first time (*Isik et al., 2014*; *Carlson et al., 2011*; *Thorpe et al., 1996*). Overlap between perception and imagery in low-level visual cortex depends on the imagery task and experienced vividness (*Lee et al., 2012*; *Albers et al., 2013*; *Kosslyn and Thompson, 2003*). However, we did not observe a relationship between overlap at this time point and imagery vividness (*Figure 3—figure supplement 1*). This absence of early overlap seems to imply that, even though early visual cortex has been implicated in visual imagery, there is no consistent overlap between imagery and early perceptual processing. One explanation for this discrepancy is that representations in low-level visual areas first have to be sharpened by feedback connections (*Kok et al., 2012*) before they have a format that is accessible by top-down imagery. Alternatively, early perceptual activity during imagery may be more brief and variable over time than high-level activation, leading to a cancelling out when averaging over trials.

The large temporal generalization at the onset of imagery might have been partly due to the nature of the task we used. Here, the start of imagery was based on the association between the cue and the items held in working memory, which could have led to an instantaneous initiation of the visual representation after the cue. It could be the case that visual representations are activated differently if imagery is operationalized as a more constructive process. A previous study has already showed that cueing imagery from long-term memory leads to less neural overlap with perception compared to cueing imagery from working memory (*Ishai et al., 2002*). Perhaps the temporal dynamics of imagery from long-term memory are also different. It could be that if perceptual details have to be filled in from long-term memory, low-level areas are activated later, resulting in a more diagonal generalization pattern. Future studies comparing temporal generalization during imagery from short- and long-term memory are necessary to investigate this further.

An alternative explanation for the broad temporal generalization during imagery is that, compared to perception, imagery is less time-locked. If the process during imagery is shifted in time between trials, averaging over trials per time point would obscure the underlying temporal dynamics. This temporal uncertainty will have a different effect on different underlying processes. For example, if the underlying process would be entirely sequential, meaning that independent representations are activated after each other (like at the start of perception), temporal shifting between trials would smear out this sequence over time. This would result in a broader diagonal pattern, where the width of the diagonal is proportional to the temporal shift between trials. This means that the broad temporal generalization that we observed during imagery could represent a sequential process if there were temporal shifts of more than a second on average between trials. Alternatively, the underlying process could be only sequential in the onset, such that different areas become active after each other, but remain active throughout time (or get reactivated later in time). In this case, temporal shifts between trials that are proportional to the difference in onset between the two processes would entirely obscure this dynamic. Note that this would mean that the start of significant imagery classification did not reflect the actual imagery onset, but the first point in time that the representations were consistent enough over trials to lead to above chance classification. Stimulus representations could actually be initiated before 350 ms after cue onset, but we would be unable to classify them at these early time points due to jitter in the onset. We cannot confidently rule out temporal uncertainty as an explanation for the broad temporal generalization at the onset of imagery. To fully resolve this issue, future research should systematically explore the effect of temporal

uncertainty on different underlying processes and analysis tools need to be developed that can account for variation in temporal dynamics between trials.

We observed clear overlap between imagery and perceptual processing around 160 ms after stimulus onset. The perceptual representation at this time likely reflects the face-specific N170. This component has been shown to be involved in face processing and appears between 130 to 170 ms after stimulus onset (*Bentin et al., 1996*; *Halgren et al., 2000*), which corresponds well with the timing of overlap with imagery reported here. The sources of the N170 are thought to be face selective areas in the ventral stream (*Deffke et al., 2007*; *Henson et al., 2009*), which also corresponds to the location of our source reconstruction at this time point. A previous study showed an increase of the N170 after imagery of a face, indicating that imagery activates similar representations as the N170 (*Ganis and Schendan, 2008*). Here we confirm that idea and show that N170 representations are active during imagery throughout time. Furthermore, this time also showed long temporal generalization within perception, indicating that the N170 representations also remain active over time during perception.

The lack of generalization between imagery and perceptual processing around 210 ms after stimulus onset was unexpected. This time window also showed an increase in temporal specificity during perception, indicating rapidly changing representations. One possible interpretation is that around this time feedback from higher areas arrives in low-level visual cortex (*Koivisto et al., 2011*; *Roelfsema et al., 1998*). If low-level representations are indeed more transient, this would explain the decrease in consistent generalization. Another possibility is that processing at this time reflects an unstable combination of feedback and feedforward processes, which is resolved around 300 ms when representations become more generalized and again start to generalize to imagery. In line with this idea, processing from 300 ms after stimulus onset has been associated with percept stabilization (*Bachmann, 2006*; *Carlson et al., 2013*; *Kaiser et al., 2016*). Future studies looking at changes in effective connectivity over time are needed to dissociate these interpretations.

Surprisingly, we did not observe any influences of experienced imagery vividness on the overlap between perception and imagery over time (Fig. S2). One explanation for this is that we used whole-brain signals for decoding whereas the relationship between overlap and vividness has only been found for a specific set of brain regions (*Lee et al., 2012*; *Dijkstra et al., 2017a*). Furthermore, if there is indeed strong temporal variability during imagery this would make it difficult to find any effect of vividness on specific time points. More studies on imagery vividness using MEG are necessary to explore this matter further.

In conclusion, our findings suggest that, in contrast to perception, at the onset of imagery the entire visual representation is activated at once. This might partly be caused by the nature of our task, since visual representations were already present in working memory at the onset of imagery. However, more research is needed to fully explore the contribution of temporal uncertainty between trials to this broad temporal generalization. Furthermore, imagery consistently overlapped with perceptual processing around 160 ms and from 300 ms onwards. This reveals the temporal counterpart of the neural overlap between imagery and perception. The overlap around 160 ms points towards a re-activation of the N170 during imagery, whereas the lack of overlap with perceptual processes before 130 ms indicates that either imagery does not rely on early perceptual representations, or that these representations are more transient and variable over time. Together, these findings reveal important new insights into the neural mechanisms of visual imagery and its relation to perception.

## Materials and methods

### Participants

We assumed a medium effect size (d = 0.6) which, to reach a power of 0.8, required twenty four participants. To take into account drop-out, thirty human volunteers with normal or corrected-to-normal vision gave written informed consent and participated in the study. Five participants were excluded: two because of movement in the scanner (movement exceeded 15 mm), two due to incorrect execution of the task (less than 50% correct on the catch trials, as described below) and one due to technical problems. 25 participants (mean age 28.6, SD = 7.62) remained for the final analysis. The study was approved by the local ethics committee and conducted according to the corresponding ethical guidelines (CMO Arnhem-Nijmegen).

## Procedure and experimental design

Prior to scanning, participants were asked to fill in the Vividness of Visual Imagery Questionnaire (VVIQ): a 16-item questionnaire in which participants indicate their imagery vividness for a number of scenarios on a 5-point scale (*Marks, 1973*). The VVIQ has been used in many imagery studies and is a well-validated measure of general imagery ability (*Lee et al., 2012*; *Albers et al., 2013*; *Dijkstra et al., 2017a*; *Cui et al., 2007*). The score was summarized in a total between 16 and 80 (low score indicates high vividness). Subsequently, the participants practiced the experimental task for ten trials outside the scanner, after which they were given the opportunity to ask clarification questions about the task paradigm. If they had difficulty with the task, they could practice a second time with ten different trials.

The experimental task is depicted in *Figure 1*. We adapted a retro-cue paradigm in which the cue was orthogonalized with respect to the stimulus identity (*Harrison and Tong, 2009*). Participants were shown two images after each other, a face and a house, or a house and a face, followed by a retro-cue indicating which of the images had to be imagined. After the cue, a frame was shown in which the participants had to imagine the cued stimulus as vividly as possible. After this, they had to indicate their experienced imagery vividness by moving a bar on a continuous scale. The size of the scale together with the screen resolution led to discretized vividness values between $-150$ and $+150$. To prevent preparation of a motor response during imagery, which side (left or right) indicated high vividness, was pseudo-randomized over trials.

The face stimuli were adapted from the multiracial face database (courtesy of Michael J Tarr, Center for the Neural Basis of Cognition and Department of Psychology, Carnegie Mellon University (Pittsburgh, Pennsylvania), http://www.tarrlab.org. Funding provided by NSF award 0339122). The house stimuli were adapted from the Pasedena houses database collected by Helle and Perona (California Institute of Technology, Pasadena, California). We chose faces and houses because these two categories elicit very different neural responses throughout the visual hierarchy, during both perception and imagery (*Ishai et al., 2000*; *Epstein et al., 2003*; *Kanwisher et al., 1997*), and are therefore expected to allow for high-fidelity tracking of their corresponding neural representations.

To ensure that participants were imagining the stimuli with great visual detail, both categories contained eight exemplars, and on 7% of the trials the participants had to indicate which of four exemplars they imagined (*Figure 1*, Catch trial). The exemplars were chosen to be highly similar in terms of low-level features to minimize within-class variability and increase between-class classification performance. We instructed participants to focus on vividness and not on correctness of the stimulus, to motivate them to generate a mental image including all visual features of the stimulus. The stimuli encompassed $2.7 \times 2.7$ visual degrees. A fixation bull's-eye with a diameter of 0.1 visual degree was on screen throughout the trial, except during the vividness rating. In total, there were 240 trials, 120 per category, divided in ten blocks of 24 trials, lasting about 5 min each. After every block, the participant had the possibility to take a break.

## MEG recording and preprocessing

Data were recorded at 1200 Hz using a 275-channel MEG system with axial gradiometers (VSM/CTF Systems, Coquitlam, BC, Canada). For technical reasons, data from five sensors (MRF66, MLC11, MLC32, MLF62, MLO33) were not recorded. Subjects were seated upright in a magnetically shielded room. Head position was measured using three coils: one in each ear and one on the nasion. Throughout the experiment head motion was monitored using a real-time head localizer (*Stolk et al., 2013*). If necessary, the experimenter instructed the participant back to the initial head position during the breaks. This way, head movement was kept below 8 mm in most participants. Furthermore, both horizontal and vertical electro-oculograms (EOGs), as well as an electrocardiogram (ECG) were recorded for subsequent offline removal of eye- and heart-related artifacts. Eye position and pupil size were also measured for control analyses using an Eye Link 1000 Eye tracker (SR Research).

Data were analyzed with MATLAB version R2017a and FieldTrip (*Oostenveld et al., 2011*) (RRID: SCR_004849). Per trial, three events were defined. The first event was defined as 200 ms prior to onset of the first image until 200 ms after the offset of the first image. The second event was defined similarly for the second image. Further analyses focused only on the first event, because the neural

response to the second image is contaminated by the neural response to the first image. Finally, the third event was defined as 200 ms prior to the onset of the retro-cue until 500 ms after the offset of the imagery frame. As a baseline correction, for each event, the activity during 300 ms from the onset of the initial fixation of that trial was averaged per channel and subtracted from the corresponding signals.

The data were down-sampled to 300 Hz to reduce memory and CPU load. Line noise at 50 Hz was removed from the data using a DFT notch filter. To identify artifacts, the variance of each trial was calculated. Trials with high variance were visually inspected and removed if they contained excessive artifacts. After artifact rejection, on average 108 perception face trials (±11), 107 perception house trials (±12) and 105 imagery face trials (±16) and 106 imagery house trials (±13) remained for analysis. To remove eye movement and heart rate artifacts, independent components of the MEG data were calculated and correlated with the EOG and ECG signals. Components with high correlations were manually inspected before removal. The eye tracker data was cleaned separately by inspecting trials with high variance and removing them if they contained blinks or other excessive artifacts.

## Decoding analyses

To track the neural representations within perception and imagery, we decoded the stimulus category from the preprocessed MEG signals during the first stimulus and after the retro-cue for every time point. To improve the signal-to-noise ratio, prior to classification, the data were averaged over a window of 30 ms centered on the time point of interest. We used a linear discriminant analysis (LDA) classifier with the activity from the 270 MEG sensors as features (see *Mostert et al., 2015* for more details). A 5-fold cross-validation procedure was implemented where for each fold the classifier was trained on 80% of the trials and tested on the other 20%. To prevent a potential bias in the classifier, the number of trials per class was balanced per fold by randomly removing trials from the class with the most trials until the trial numbers were equal between the classes.

## Generalization across time and conditions

By training a classifier on one time point and then testing it on other time points, we were able to investigate the stability of neural representations over time. The resulting temporal generalization pattern gives information about the underlying processing dynamics. For instance, a mostly diagonal pattern reflects sequential processing of specific representations, whereas generalization from one time point towards another reflects recurrent or sustained activity of a particular process (*King and Dehaene, 2014*). Here, we performed temporal generalization analyses during perception and during imagery to investigate the dynamics of the neural representations. Furthermore, to quantify the extent to which the representation at a given time point $t$ was specific to that time point, we tested whether a classifier trained at time $t$ and tested at time $t$ (i.e. diagonal decoding) had a higher accuracy than a classifier trained at time $t'$ and tested at time $t$ (i.e. generalization). This shows whether there is more information at time $t$ than can be extracted by the decoder $t'$ (*Astrand et al., 2015*; *King et al., 2014*). We subsequently calculated, for each training time point, the proportion of testing time points that were significantly lower than the diagonal decoding, giving a measure of specificity for each time point. To avoid overestimating the specificity, we only considered the time window during which the diagonal classifiers were significantly above chance.

To investigate the overlap in neural representations between perception and imagery, a similar approach can be used. Here, we trained a classifier on different time points during perception and tested it on different time points during imagery and vice versa. This analysis shows when neural activity during perception contains information that can be used to dissociate mental representations during imagery and vice versa - that is which time points show representational overlap. For both the temporal generalization as well as the across condition generalization analysis, we also applied cross-validation to avoid overestimating generalization due to autocorrelation in the signals.

It has been shown that representational overlap between imagery and perception, as measured by fMRI, is related to experienced imagery vividness (*Lee et al., 2012*; *Albers et al., 2013*; *Dijkstra et al., 2017a*). To investigate this in the current study, we performed a median split on the averaged vividness across trials on the group level, which yielded a high vividness ($N$ = 12, vividness: 71.64 ± 12.44) and a low vividness ($N$ = 12, vividness: 27.25 ± 17.69) group. We produced the

accuracy maps for all previous analyses separately for the two groups and compared the decoding accuracies of the two groups using cluster based permutation testing (see Statistical testing).

## Statistical testing

Decoding accuracy was tested against chance using two-tailed cluster-based permutation testing with 1000 permutations (*Maris and Oostenveld, 2007*). In the first step of each permutation, clusters were defined by adjacent points that crossed a threshold of $p<0.05$. The t-values were summed within each cluster, but separately for positive and negative clusters, and the largest of these were included in the permutation distributions. A cluster in the true data was considered significant if its p-value was less than 0.05 based on the permutations. Correlations with vividness were tested against zero on the group level using the same procedure.

## Source localization

In order to identify the brain areas that were involved in making the dissociation between faces and houses during perception and imagery, we performed source reconstruction. In the case of LDA classifiers, the spatial pattern that underlies the classification reduces to the difference in magnetic fields between the two conditions (see *Haufe et al., 2014*). Therefore, to infer the contributing brain areas, we performed source analysis on the difference ERF between the two conditions.

For this purpose, T1-weighted structural MRI images were acquired using a Siemens 3T whole body scanner. Vitamin E markers in both ears indicated the locations of the head coils during the MEG measurements. The location of the fiducial at the nasion was estimated based on the anatomy of the ridge of the nose. The volume conduction model was created based on a single shell model of the inner surface of the skull. The source model was based on a reconstruction of the cortical surface created for each participant using FreeSurfer's anatomical volumetric processing pipeline (RRID: SCR_001847). MNE-suite (Version 2.7.0; RRID: SCR_005972) was subsequently used to infer the subject-specific source locations from the surface reconstruction. The resulting head model and source locations were co-registered to the MEG sensors.

The lead fields were rank reduced for each grid point by removing the sensitivity to the direction perpendicular to the surface of the volume conduction model. Source activity was obtained by estimating linearly constrained minimum variance (LCMV) spatial filters (*Van Veen et al., 1997*). The data covariance was calculated over the interval of 50 ms to 1 s after stimulus onset for perception and over the entire segment for imagery. The data covariance was subsequently regularized using shrinkage with a regularization parameter of 0.01 (as described in *Manahova et al., 2017*). These filters were then applied to the axial gradiometer data, resulting in an estimated two-dimensional dipole moment for each grid point over time. For imagery, the data were low-pass filtered at 30 Hz prior to source analysis to increase signal to noise ratio.

To facilitate interpretation and visualization, we reduced the two-dimensional dipole moments to a scalar value by taking the norm of the vector. This value reflects the degree to which a given source location contributes to activity measured at the sensor level. However, the norm is always a positive value and will therefore, due to noise, suffer from a positivity bias. To counter this bias, we employed a permutation procedure in order to estimate this bias. Specifically, in each permutation, the sign of half of the trials were flipped before averaging and projecting to source space. This way, we cancelled out the systematic stimulus-related part of the signal, leaving only the noise. Reducing this value by taking the norm thus provides an estimate of the positivity bias. This procedure was repeated 1000 times, resulting in a distribution of the noise. We took the mean of this distribution as providing the most likely estimate of the noise and subtracted this from the true, squared source signal. Furthermore, this estimate provides a direct estimate of the artificial amplification factor due to the depth bias. Hence, we also divided the data by the noise estimate to obtain a quantity that allowed visualization across cortical topography. For full details, see *Manahova et al. (2017)*.

For each subject, the surface-based source points were divided into 74 atlas regions as extracted by FreeSurfer on the basis of the subject-specific anatomy (*Destrieux et al., 2010*). To enable group-level estimates, the activation per atlas region was averaged over grid points for each participant. Group-level activations were then calculated by averaging the activity over participants per atlas region (*van de Nieuwenhuijzen et al., 2016*).

## Acknowledgements

The authors would like to thank Marius Peelen for his comments on an earlier version of the manuscript and Jean-Rémi King for his suggestions for analyses.

## Additional information

### Competing interests

Floris P de Lange: Reviewing Editor, *eLife*. The other authors declare that no competing interests exist.

### Funding

| Funder | Grant reference number | Author |
|---|---|---|
| Nederlandse Organisatie voor Wetenschappelijk Onderzoek | 639.072.513 | Sander Bosch<br>Marcel AJ van Gerven |
| Horizon 2020 Framework Programme | 678286 | Floris P de Lange |
| Nederlandse Organisatie voor Wetenschappelijk Onderzoek | 452-13-016 | Floris P de Lange |
| Nederlandse Organisatie voor Wetenschappelijk Onderzoek | 406-13-001 | Pim Mostert |

The funders had no role in study design, data collection and interpretation, or the decision to submit the work for publication.

### Author contributions

Nadine Dijkstra, Conceptualization, Data curation, Formal analysis, Funding acquisition, Investigation, Visualization, Methodology, Writing—original draft, Project administration, Writing—review and editing; Pim Mostert, Conceptualization, Software, Formal analysis, Investigation, Visualization, Methodology, Writing—review and editing; Floris P de Lange, Conceptualization, Investigation, Writing—review and editing; Sander Bosch, Conceptualization, Data curation, Supervision, Investigation, Project administration, Writing—review and editing; Marcel AJ van Gerven, Conceptualization, Supervision, Methodology, Project administration, Writing—review and editing

### Author ORCIDs

Nadine Dijkstra (iD) http://orcid.org/0000-0003-1423-9277

### Ethics

Human subjects: Informed consent and consent to publish was obtained from all participants and the experiment was conducted according to the ethical guidelines provided by the CMO Arnhem-Nijmegen.

### Decision letter and Author response

Decision letter https://doi.org/10.7554/eLife.33904.020
Author response https://doi.org/10.7554/eLife.33904.021

## Additional files

### Supplementary files

• Transparent reporting form
DOI: https://doi.org/10.7554/eLife.33904.015

## Data availability

The data discussed in the manuscript are available at the Donders Repository (https://data.donders. ru.nl/collections/shared/di.dcc.DSC_2017.00072_245?2). To download the data, the user has to agree with the accompanying Data Use Agreement (DUA), which states, a.o. that the user will not attempt to identify participants from the data and will acknowledge the origin of the data in any resulting publication. The DUA can be signed and the data can be downloaded by clicking on 'Request access' and logging in using either an institutional or a social account (e.g. Google+, Twitter, LinkedIn).

The following dataset was generated:

| Author(s) | Year | Dataset title | Dataset URL | Database, license, and accessibility information |
|---|---|---|---|---|
| Nadine Dijkstra, Pim Mostert, Sander Bosch, Floris P de Lange, Marcel AJ van Gerven | 2018 | Temporal dynamics of visual imagery and perception | https://data.donders.ru.nl/collections/shared/di.dcc.DSC_2017.00072_245?2 | Publicly available at Donders Repository |

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
