## [Decision Letter]

Thank you for submitting your article "Differential temporal dynamics during visual imagery and perception" for consideration by *eLife*. Your article has been reviewed by three peer reviewers, and the evaluation has been overseen by a Reviewing Editor and Sabine Kastner as the Senior Editor. The following individual involved in review of your submission has agreed to reveal his identity: Nicholas Edward Myers (Reviewer #2).

The reviewers have discussed the reviews with one another and the Reviewing Editor has drafted this decision to help you prepare a revised submission.

Summary:

In this paper by Dijkstra et al., entitled "Differential temporal dynamics during visual imagery and perception", the authors use machine learning methods (decoding) to compare the temporal dynamics of whole brain MEG signals during the perception of faces vs. houses on the one hand and the visual imagery of these same visual categories (faces vs. houses). Previous studies have described how much of the visual perception network is also recruited during visual imagery and how this recruitment depends on the specifics of the task design. The major novelty of this study is to address the temporal aspect of information flow in the relevant cortical networks. The methods are state of the art methods and the reported observations are straightforward.

The reviewers and editors thought that greater attempts need to be made to make a point with respect to our understanding of imagery. At the moment, this paper is methodologically sound, carefully done, and well written. However, we think it has a very descriptive character, presenting a result that is not entirely surprising (imagery decoding is later, weaker, and more all over the place) and offering two different interpretations. That's a bit unsatisfying. To also make a stronger theoretical contribution, we think the authors should make a greater effort to disentangle these hypotheses (and in addition, a third interpretation that came up during the review process).

Essential revisions:

1) How does the cross-temporal decoding of faces vs. houses compare to the cross-temporal decoding of faces (resp. houses) in each task. In other words, for each task, how does the cross-category signal variability compare to the within category signal variability in each condition. The reviewers and editors acknowledge that the authors go through a random permutation procedure to identify data-based chance level. Our question pertains to a better understanding of the neural bases of imagery: could it be that overall lower accuracies during imagery is due to a higher variability in within category information? This is a way to address whether the decay in decoding accuracy during imagery is due to weaker representations or increased representational variability.

2) When presenting the temporal overlap between perception and imagery analysis, and Figure 3, the authors mention that the reported results describing that a classifier trained on perception can extract information from imagery data much earlier than a classifier trained on imagery data, might be due to the fact that more trials are available to the trained. This is a crucial point. We strongly recommend that the authors down sample their perception data to make this analysis (both that of Figure 3A and 3C) more comparable to the data presented in Figure 2. In this absence of this control, it is difficult to interpret the presented result.

3) An important additional analysis that needs to be added is the analysis presented in Figure 2C on the high vividness and low vividness trials separately (on matched samples sizes). One expects there to have an overall lower decoding accuracy on low vividness trials. This analysis we expected to directly relate imagery decoding accuracy with the subjective report of imagery vividness. This would a be prior analysis to the one presented in Figure S2, addressing whether more overlap of imagery representation with perceptual representations correlated with higher vividness in imagery subjective experience (further discussed in the next point).

4) The analysis presented in Figure S2, correlating perception/imagery representational overlap with imagery vividness is a crucial analysis to strengthen the scope of the present paper. It is based onto a LDA, thus assuming a linear relationship between imagery vividness and similarity between perceptual and imagery representations. One can expect other possible relationships between the two: e.g. a one or nothing mapping or a sigmoid type of mapping. We recommend to the authors to explore this differently. One way of doing it would be to perform to separate analysis as in figure 3 for high vividness imagery trials and low vividness imagery trials. One expects a difference to support the subject experience of imagery, though this difference might be localized to some cortical sources (here, the authors will have to make sure to use the same number of trials from training and testing and for the high and low vividness comparison).

5) The conclusion states that the 'findings show that imagery is characterized by a different temporal profile than perception.' This conclusion doesn't sound entirely surprising, and makes us think that the authors may be selling themselves short. We think it would be important to highlight what the results actually say about the neural basis of imagery.

6) Furthermore, it might help the Discussion if the authors speculated how specific their results are to the task they chose, rather than imagery per se. In their task, imagery is always preceded by presentation of the exact stimulus to be imagined (along with a not-to-be imagined control stimulus), which means that participants will still have a lot of information about the imagined stimulus in visual working memory. This would not be the case if, for example, participants were suddenly prompted to imagine a novel object. Could this partially account for the good cross-generalization between perception and imagery? What if there had been a control condition requiring memory but no imagery? Without such a control, how much of their findings may be attributed to visual working memory, rather than imagery?

7) The clear decoding peak around 160 ms seems like it could be related to the N170 or M170 component. Since this component is so well-studied in terms of its role in face processing and its cortical sources in the ventral stream, it seems warranted to discuss this a bit more. Does the fact that the 160 ms peak cross-generalizes well across time and from perception to imagery indicate that face- or house-selective areas in the ventral stream are active at this time and then maintain activation later on, especially during imagery?

8) The authors raise two different explanations for their results, e.g., in the Abstract: "These results indicate that during imagery either the complete representation is activated at once and does not include low-level visual areas, or the order in which visual features are activated is less fixed and more flexible than during perception." However, it is possible that there is a probably less exciting, yet more parsimonious explanation for the pattern of results.

Perception and imagery could just evoke the same processing cascade (which the authors just dismiss us "unlikely"; main text, third paragraph), with two restrictions: (a) Imagery might not "start" as early as perception, thus not involving the earliest stages of perceptual processing. Thus, the onset of imagery decoding is expected to be later than the onset of perception decoding. (b) Visual imagery is initiated in a top-down way; in contrast to the clear onset of a perceptual event, this initiation of imagery may vary substantially between trials (e.g., by an accumulation of temporal noise). Thus, the onset of imagery decoding is expected to have a smoother rise (obscuring the initial dynamics of the signal), and "temporal generalization" would increase a lot (as the evoked responses for single trials are relatively shifted across time).

Importantly, this explanation would neither suggest that in imagery complete representations are activated at once, nor that the order of processing steps is somehow altered as compared to perception. Note how also the "dip" in cross decoding at 210 ms is explained by this account, without evoking a more complicated explanation: If the neural representation at 210 ms after onset of the event is just very distinct (for whatever reason), the same signal scattered in time from trial to trial would impair cross-decoding specifically for this very time point (where temporal uncertainty hurts a lot). Do the authors think that their data are consistent with this alternative explanation, or are their data points that refute this account?

9) To dissociate the temporal uncertainty explanation and the explanations given by the authors, I would strongly suggest additional analyses that try to estimate – and potentially correct for – the temporal uncertainty in imagery-related activation. One such analysis could try to align the imagery trials based on the within-imagery decoding. For example, the authors could perform a leave-one-trial-out analysis where they train a classifier on all trials, and test the classifier on the remaining trial, while shifting the left-out trial in time until the maximum decoding accuracy can be reached (this analysis should probably be done by randomly subsampling a smaller amount of trials as the testing set to reach more reliable estimates). Then for each trial's imagery period, this optimal temporal shift can be used to re-epoch the signal. If the processing sequence is similar and just suffers from temporal scattering, this procedure should significantly improve the cross-decoding accuracy while decreasing temporal generalization.

---

## [Author Response]

[…] The reviewers and editors thought that greater attempts need to be made to make a point with respect to our understanding of imagery. At the moment, this paper is methodologically sound, carefully done, and well written. However, we think it has a very descriptive character, presenting a result that is not entirely surprising (imagery decoding is later, weaker, and more all over the place) and offering two different interpretations. That's a bit unsatisfying. To also make a stronger theoretical contribution, we think the authors should make a greater effort to disentangle these hypotheses (and in addition, a third interpretation that came up during the review process).

We agree with the reviewers and editors that the paper would benefit from more focus on the neural mechanisms of imagery.

Specifically, we have:

- Performed within-category decoding for both perception and imagery (issue 1). We have added an explanation for the difference in decoding accuracy between perception and imagery in the manuscript;

- Performed additional analyses on vividness (issues 3 and 4) and have added the results to the manuscript;

- Reworked the conclusion to better highlight the contribution of our findings to the field (issue 5);

- Added an explanation of the temporal generalization results with respect to task specificity (issue 6);

- Connected the temporal cross-decoding results to the N170 (issue 7);

- Performed additional analyses to try to accommodate the temporal uncertainty within imagery (issue 8 and 9) and added more discussion on this issue in the manuscript.

Essential revisions:1) How does the cross-temporal decoding of faces vs. houses compare to the cross-temporal decoding of faces (resp. houses) in each task. In other words, for each task, how does the cross-category signal variability compare to the within category signal variability in each condition. The reviewers and editors acknowledge that the authors go through a random permutation procedure to identify data-based chance level. Our question pertains to a better understanding of the neural bases of imagery: could it be that overall lower accuracies during imagery is due to a higher variability in within category information? This is a way to address whether the decay in decoding accuracy during imagery is due to weaker representations or increased representational variability.

We agree with the reviewers that exploring the relationship between within-category and between-category decoding during both imagery and perception is interesting. However, we optimized the design of the current experiment for between-category decoding. Given that imagery produces a relatively noisy signal, we attempted to maximize our signal-to-noise ratio (SNR).

Ideally, there would be only one stimulus per category, which is maximally different from the stimulus in the other category, and we would obtain as many repetitions as possible. However, we also needed participants to stay engaged in the task and motivate them to form detailed mental images. Therefore, they performed a more challenging exemplar identification task.

The stimuli in this task consisted of 8 exemplars per stimulus category that were as similar as possible (faces: all female, dark, long hair; houses: all a slanted roof, and relatively square), but the two stimulus sets (faces versus houses) were very different. We had a maximum of 120 trials per category (before trial rejection), which gave enough power for low but reliable above chance classification within imagery. In contrast, for instance-based decoding we would have 120/8 = a maximum of 15 trials per class.

To illustrate this issue fully, we have run pairwise classifiers on all combinations of stimuli (16 x 16; 8 per class) for the peak decoding latency for perception (~160 ms) and for imagery (~1 s). As can be seen (Author response image 1, bottom plots), the decoding accuracies for between-category classification are considerably lower than before (perception was at 90% and is now at 70%, imagery was at 60% and is now at ~50%), probably due to the decrease in the number of trials for each comparison. Furthermore, within-class classification performance is slightly above chance for perception, but entirely at chance for imagery.

**Author response image 1. respfig1:** Within category decoding accuracy for perception (left) and imagery (right). At the top the pairwise decoding accuracies for the different stimuli are shown where F are the face stimuli and H are the house stimuli. At the bottom, the averaged accuracies for the different comparisons (within faces, within houses, between categories) are shown as a distribution over participants.

Considering the large drop in between-category decoding accuracy with fewer trials, it is impossible to fully investigate this question with our data. However, the fact that within-class decoding was above chance in perception and completely at chance for imagery suggests that there is no increase in within-class variability in imagery compared to perception. We believe that the overall lower accuracies during imagery compared to perception are due to an increase in between-trial variability during imagery due to its cognitive nature: perception is a more automatic process that will proceed in a similar way in every instance, whereas imagery is a demanding process that relies on a combination of different cognitive abilities (working memory retrieval, visualization, percept stabilization). There will likely be larger variation between trials for a cognitively more complex process.

We agree with the reviewers that more explanation for this difference in decoding accuracy is warranted and have added the following to the text:

“The generation of a visual representation from a cue thus seems to take longer than the activation via bottom-up sensory input. […] This is also observed in fMRI decoding studies (Reddy et al., 2010; Lee et al., 2012) and is probably due to the fact that imagery is a less automatic process than perception, which leads to higher between trial variation in neural activations.”

2) When presenting the temporal overlap between perception and imagery analysis, and Figure 3, the authors mention that the reported results describing that a classifier trained on perception can extract information from imagery data much earlier than a classifier trained on imagery data, might be due to the fact that more trials are available to the trained. This is a crucial point. We strongly recommend that the authors down sample their perception data to make this analysis (both that of Figure 3A and 3C) more comparable to the data presented in Figure 2. In this absence of this control, it is difficult to interpret the presented result.

We thank the reviewers for pointing out this issue. These results were written based on an earlier version of the analysis in which we did not yet apply cross-validation to the cross-decoding analysis, which meant that there were more data in this analysis compared to the within-condition decoding. However, we realized that this could lead to false positives due to autocorrelations in the signal. Therefore, for the final cross-decoding analyses which are reported here, we did apply cross-validation, leading to the same amount of training data as for the within-imagery (and within-perception) analyses. Therefore, the earlier decoding here cannot be explained by a difference in amount of training data.

To clarify this point, we have added the following sentence to the Materials and methods section “Generalization across time and conditions”:

“For both the temporal generalization as well as the across condition generalization analysis, we also applied cross-validation to avoid overestimating generalization due to autocorrelation in the signals.”

Even though the earlier decoding of imagery based on perception could not be due to a difference in amount of training data, it could still reflect differences in SNR between the two analyses. As can be seen from the higher decoding accuracy during perception (see previous point), perception has a higher SNR. Therefore, training (or testing) on perception will result in increased decoding performance. To clarify this point, we have added the following to the text:

“This is earlier than classification within imagery, which started at 540 ms after cue onset (Figure 2A, right). Considering the increased decoding accuracy during perception compared to imagery (Figure 2A, left versus right), this difference might reflect an increase in signal-to-noise ratio (SNR) by training on perception compared to imagery.”

3) An important additional analysis that needs to be added is the analysis presented in Figure 2C on the high vividness and low vividness trials separately (on matched samples sizes). One expects there to have an overall lower decoding accuracy on low vividness trials. This analysis we expected to directly relate imagery decoding accuracy with the subjective report of imagery vividness. This would a be prior analysis to the one presented in Figure S2, addressing whether more overlap of imagery representation with perceptual representations correlated with higher vividness in imagery subjective experience (further discussed in the next point).

We agree with the reviewer that it is important to explore the effects of vividness on the imagery decoding. We have explored the effects of vividness on decoding accuracy more thoroughly and reported the results (see our response to issue 4 below).

4) The analysis presented in Figure S2, correlating perception/imagery representational overlap with imagery vividness is a crucial analysis to strengthen the scope of the present paper. It is based onto a LDA, thus assuming a linear relationship between imagery vividness and similarity between perceptual and imagery representations. One can expect other possible relationships between the two: e.g. a one or nothing mapping or a sigmoid type of mapping. We recommend to the authors to explore this differently. One way of doing it would be to perform to separate analysis as in figure 3 for high vividness imagery trials and low vividness imagery trials. One expects a difference to support the subject experience of imagery, though this difference might be localized to some cortical sources (here, the authors will have to make sure to use the same number of trials from training and testing and for the high and low vividness comparison).

We agree with the reviewers, and have looked at decoding accuracy for high and low vividness separately.

When performing these analyses, we realized that prior to imagery the signal corresponding to the second image will be present more strongly in the data than the signal corresponding to the first image (partly due to signal bleed-in but also due to the well-known recency effect during working memory, see i.e. Morrison, Conway and Chein, 2014). Since we used a two-class design in which the class of the imagined stimulus either corresponds to the second image (in cue 2 trials) or to the other class (in cue 1 trials), this means that this bleed-in/recency *increases* the accuracy for cue 2 trials and *decreases* the accuracy for cue 1 trials, irrespective of any imagery. This effect will likely influence the accuracy throughout the imagery trial. Usually, one averages over all trials and thereby cancels out this effect. However, there are more high vividness cue 2 trials than cue 1 trials (probably due to the recency effect). This means that effects of vividness between trials within participants will likely be influenced by this bleed-in. Therefore, we decided to instead only look at group-level effects of vividness.

This also means that the analysis that we proposed earlier, in which we correlate the LDA distance to the vividness, might have been influenced by recency and/or bleed-in. We decided to remove that analysis from the paper and replace it by the group-level analysis.

We performed a median split on the averaged vividness across trials on the group level, which yielded a high vividness (*N* = 12, vividness: 71.64 ± 12.44) and a low vividness (*N* = 12, vividness: 27.25 ± 17.69) group. We produced the accuracy maps for all previous analyses and plotted the difference between high and low vividness in supplementary figures of the main figures (see Figure 2—figure supplement 3 and Figure 3—figure supplement 1).

There were no significant differences in any of the accuracies after correction for multiple comparisons. However, we do see that the effect generally goes in the expected direction: there is on average a higher decoding accuracy during imagery for the high vividness group compared to the low vividness group (Figure 2C and D). Surprisingly, this also seems to be the case for the accuracy during perception (panels A and B). This might reflect a general attention modulation or a working memory encoding effect. Furthermore, there seems to be an interesting pattern present in the cross-decoding matrix, especially when training on imagery (Figure 3B): overlap around 100 ms and around 210 ms during perception (x-axis) seems to be higher in the high vividness group. We have added these results to the paper. It is unfortunate that we do not find any significant effects of vividness. However, as we already mentioned in the Discussion, influences of vividness have only been found in very specific brain areas. It is quite possible that these may become negligible when looking at a whole brain signal. Furthermore, if the signal suffered from temporal uncertainty (see issues 8 and 9), then it would be even harder to find a consistent effect of vividness at specific time points.

5) The conclusion states that the 'findings show that imagery is characterized by a different temporal profile than perception.' This conclusion doesn't sound entirely surprising, and makes us think that the authors may be selling themselves short. We think it would be important to highlight what the results actually say about the neural basis of imagery.

We agree that we can provide a more explicit interpretation of the neural mechanisms of imagery. Several other reviewer comments relate to this issue and we have tried to accommodate all of them. Furthermore, we extended the emphasis of the paper also to the overlap in time between perception and imagery. We have reworked the Introduction to take this into account as follows:

“However, the temporal dynamics of visual imagery remain unclear. During imagery, there is no bottom-up sensory input. […] Second, we investigated the temporal overlap by exploring which time points during perception generalized to imagery.”

Furthermore, we have adapted the conclusion in the Discussion as follows:

“In conclusion, our findings suggest that, in contrast to perception, at the onset of imagery the entire visual representation is activated at once. […] Together, these findings reveal important new insights into the neural mechanisms of visual imagery and its relation to perception.”

6) Furthermore, it might help the Discussion if the authors speculated how specific their results are to the task they chose, rather than imagery per se. In their task, imagery is always preceded by presentation of the exact stimulus to be imagined (along with a not-to-be imagined control stimulus), which means that participants will still have a lot of information about the imagined stimulus in visual working memory. This would not be the case if, for example, participants were suddenly prompted to imagine a novel object. Could this partially account for the good cross-generalization between perception and imagery? What if there had been a control condition requiring memory but no imagery? Without such a control, how much of their findings may be attributed to visual working memory, rather than imagery?

We agree with the reviewers that it is important to speculate about to what extent our results are specific to the task we used. It has previously been reported that overlap between imagery and perception is indeed lower when imagery is cued from long-term memory (Ishai et al., 2002). Therefore, cueing from long-term memory would have likely also resulted in lower cross-decoding accuracy here. However, this would have been likely been caused by a number of memory-related processes (success of previous encoding, retrieval, etc.), which were not the focus of the current study. In contrast, our aim was to increase our understanding of the neural mechanisms of visual experience in the absence of bottom-up input and to what extent they are similar to when bottom-up input is present. To do this, we wanted to maximize the similarity between perception and imagery while only varying whether or not bottom-up input was present. This is what we attempted here by using a retro-cue paradigm, in which the participant saw the to-be-imagined item only seconds before. Furthermore, other research has shown that imagery and visual working memory rely on highly similar representations in visual cortex (Albers et al., 2013; Tong, 2013; Christophel et al., 2015; Lee and Baker, 2016). Accordingly, we do not aim to dissociate imagery and visual working memory here, but rather compare the temporal dynamics underlying visual experience in the presence and absence of sensory input.

However, it is possible that the temporal generalization pattern during imagery would be different if we cued from long-term memory, which would point towards different mechanisms for imagery from long-term memory versus imagery from working memory. To further connect our findings to the specifics of the task, we have added the following paragraph to the Discussion:

“The large temporal generalization at the onset of imagery might have been partly due to the nature of the task we used. […] Future studies comparing temporal generalization during imagery from short- and long-term memory are necessary to investigate this further.”

7) The clear decoding peak around 160 ms seems like it could be related to the N170 or M170 component. Since this component is so well-studied in terms of its role in face processing and its cortical sources in the ventral stream, it seems warranted to discuss this a bit more. Does the fact that the 160 ms peak cross-generalizes well across time and from perception to imagery indicate that face- or house-selective areas in the ventral stream are active at this time and then maintain activation later on, especially during imagery?

We thank the reviewers for pointing this out to us. Our findings can indeed be linked very clearly to the N170. Accordingly, we have added the following paragraph to the Discussion:

“We observed clear overlap between imagery and perceptual processing around 160 ms after stimulus onset. [...] Furthermore, this time also showed long temporal generalization within perception, indicating that the N170 representations also remain active over time during perception.”

8) The authors raise two different explanations for their results, e.g., in the Abstract: "These results indicate that during imagery either the complete representation is activated at once and does not include low-level visual areas, or the order in which visual features are activated is less fixed and more flexible than during perception." However, it is possible that there is a probably less exciting, yet more parsimonious explanation for the pattern of results.Perception and imagery could just evoke the same processing cascade (which the authors just dismiss us "unlikely"; main text, third paragraph), with two restrictions: (a) Imagery might not "start" as early as perception, thus not involving the earliest stages of perceptual processing. Thus, the onset of imagery decoding is expected to be later than the onset of perception decoding.

This third explanation is interesting, but we feel it is not in line with our findings: if there is a clear processing cascade during imagery one would expect a more diagonal pattern for the temporal generalization (unless there is a lot of temporal uncertainty between trials, see next point). Furthermore, one would expect that later time points during perception overlap with later time points during imagery, and the same for earlier time points. Even if the earliest time point that overlaps with imagery is later than the earliest time point during perception that contains information (i.e. ‘imagery starts later’), one would still observe a temporal shift in the overlap of different time points if there really was a processing cascade.

Alternatively, if the reviewers mean that imagery does not include the earliest time points, and only activates *late* visual representations, then that would refer to the cross-decoding results. We have reworked the Abstract to clarify these different explanations. Furthermore, in the Discussion we explained this point as follows:

“This absence of early overlap seems to imply that, even though early visual cortex has been implicated in visual imagery, there is no consistent overlap between imagery and early perceptual processing. […] Alternatively, early perceptual activity during imagery may be more brief and variable over time than high-level activation, leading to a cancelling out when averaging over trials.”

(b) Visual imagery is initiated in a top-down way; in contrast to the clear onset of a perceptual event, this initiation of imagery may vary substantially between trials (e.g., by an accumulation of temporal noise). Thus, the onset of imagery decoding is expected to have a smoother rise (obscuring the initial dynamics of the signal), and "temporal generalization" would increase a lot (as the evoked responses for single trials are relatively shifted across time).Importantly, this explanation would neither suggest that in imagery complete representations are activated at once, nor that the order of processing steps is somehow altered as compared to perception. Note how also the "dip" in cross decoding at 210 ms is explained by this account, without evoking a more complicated explanation: If the neural representation at 210 ms after onset of the event is just very distinct (for whatever reason), the same signal scattered in time from trial to trial would impair cross-decoding specifically for this very time point (where temporal uncertainty hurts a lot). Do the authors think that their data are consistent with this alternative explanation, or are their data points that refute this account?

Please see our response to Issue 9.

9) To dissociate the temporal uncertainty explanation and the explanations given by the authors, I would strongly suggest additional analyses that try to estimate – and potentially correct for – the temporal uncertainty in imagery-related activation. One such analysis could try to align the imagery trials based on the within-imagery decoding. For example, the authors could perform a leave-one-trial-out analysis where they train a classifier on all trials, and test the classifier on the remaining trial, while shifting the left-out trial in time until the maximum decoding accuracy can be reached (this analysis should probably be done by randomly subsampling a smaller amount of trials as the testing set to reach more reliable estimates). Then for each trial's imagery period, this optimal temporal shift can be used to re-epoch the signal. If the processing sequence is similar and just suffers from temporal scattering, this procedure should significantly improve the cross-decoding accuracy while decreasing temporal generalization.

We agree that temporal uncertainty during imagery is a very important issue, which deserves more emphasis in this paper. We thank the reviewers for this suggestion and have run this analysis (full details can be found in Appendix A: “Temporal alignment on imagery accuracy”).

We found that the best shifts for each trial were distributed quite uniformly over time. This indicates that either the representation at imagery onset does indeed generalize broadly over time, or that there is a very wide distribution of temporal jitter across trials. If the latter were true and we did identify the onset of imagery in each trial, the resulting temporal generalization matrix should show a more diagonal pattern, indicating changing representations. This is not what we found: when training and testing on the realigned data, we still observed broad off-diagonal decoding. This seems to suggest that there is indeed broad temporal generalization, although it is possible that the decoding model on which we based the onset alignment did not reflect the imagery onset.

To further investigate this issue, we aligned the trials based on lagged correlations between the MEG signal, instead of decoding accuracy (see Appendix A: “Temporal alignment on lagged correlations”). Because this analysis does not make use of decoding accuracy for the onset shift, any observed increase in accuracy would be due to an increase in temporal alignment. We did not observe any increase in decoding accuracy after aligning the trials in this way. Furthermore, the broad off-diagonal temporal generalization remained. It is difficult to draw conclusions from this because the accuracy did not increase, indicating that perhaps the single-trial signals were too noisy to get reliable cross-correlations.

Finally, we explored the recently developed Temporally Unconstrained Decoding Analysis (TUDA; Vidaurre et al., 2018), which estimates the number of unique states that are needed to describe a neural process, while allowing variation in the exact timing of these states within each trial. We also corresponded with the first author of that paper regarding the method. Unfortunately, in its current form, TUDA is not suited to answer the questions that we have (for more info, see Appendix A: “Temporally unconstrained decoding analysis”).

In conclusion, we cannot confidently dissolve the temporal uncertainty issue with the current analysis tools. We agree that temporal uncertainty is an important alternative explanation for part of our findings. Furthermore, we feel that it is more parsimonious than our earlier suggestion that the order of low-level feature activations may be more flexible during imagery. Therefore, we have replaced that part of the Discussion with the following paragraph:

“An alternative explanation for the broad temporal generalization during imagery is that, compared to perception, imagery is less time-locked. […] To fully resolve this issue, future research should systematically explore the effect of temporal uncertainty on different underlying processes and analysis tools need to be developed that can account for variation in temporal dynamics between trials.”